# Homicide Rates and the Multiple Dimensions of Urbanization: A Longitudinal, Cross-National Analysis

**Matthew Thomas Clement [1],\*, Nathan W. Pino [1] and Jarrett Blaustein [2]**

[1]   Department of Sociology, Texas State University, San Marcos, TX 78666, USA; np11@txstate.edu
[2]   School of Social Sciences, Monash University, Clayton, VIC 3800, Australia; Jarrett.Blaustein@monash.edu
\*   Correspondence: mtc73@txstate.edu

**Abstract:** Sustainability scholars frame urbanization as a multidimensional concept with divergent environmental impacts. Through synthesizing recent quantitative studies of urbanization in criminology, we evaluated this multidimensional framework in a longitudinal, cross-national analysis of homicide rates for 217 countries between 2000 and 2015. For the analysis, we also highlighted the issue of missing data, a common concern for cross-national scholars in a variety of disciplines. While controlling for other relevant factors, we compared results from panel models that use the common technique of listwise deletion ($n = 113$) and from structural equation models (SEM) that handle missing values with full information maximum likelihood ($n = 216$). While the estimates for the control variables are non-significant in the SEM approach, the findings for the urbanization variables were robust and multidimensional. In particular, while the proportion of the population that is urban is positively related to homicide, the proportion of the population living in large cities of at least one million inhabitants is significantly and negatively related to homicide in all models. Given our focus on urbanization, we outline our contribution not only in the context of criminology but also the cross-national sustainability literature, which often uses similar variables with missing values.

**Keywords:** urbanization; homicide rates; cross-national; missing values

## 1. Introduction

Many quantitative, cross-national studies that examine the relationship between development and crime focus on homicide rates, since they are seen as a reliable measure that can gauge a country's general criminal violence rate [1]. The present study used country-level data from the World Development Indicators, and for one variable, the Standardized World Income Inequality Database, to analyze the structural predictors of homicide, with a focus on urbanization. While we controlled for several relevant factors, we focused on the association between homicide rates and urbanization for two reasons: first, because it provides us with an opportunity to revisit this relationship which has been at the center of the crime-development literature; and second, because the process of urbanization is central to understanding the dynamics of development and sustainability.

Rather than measure urbanization as a single variable, we drew on sustainability scholarship that treats urbanization as a multidimensional concept with countervailing and non-monotonic consequences [2,3]. By synthesizing recent local-level and cross-national criminological research [4–7], we evaluated this multidimensional framework in a cross-national study of homicide rates. Regardless of the way missing values were handled, the results from our analysis highlighted urbanization variables, not other factors, such as age structure or economic productivity, as robust predictors of homicide rates. Furthermore, following the approach in sustainability scholarship, we found evidence

that urbanization is multidimensional with countervailing criminological consequences. Given our focus on urbanization and that cross-national sustainability scholars use similar data sets [3], we framed our study as part of an emerging literature drawing connections between crime and sustainability [4,8].

While we used country-level data to analyze homicide rates, we acknowledge that the cross-national homicide literature as a whole suffers from data limitations, especially in terms of sample size [9]. Maximizing sample size is a pressing issue, as the variables in cross-national data sets, especially longitudinal ones, have incomplete coverage with missing values. Indeed, the issue of missing values has an impact on cross-national research in a variety of disciplines using similar data sets. For instance, in our study of homicide rates, we drew from the sustainability literature to evaluate the use of urbanization variables in cross-national studies by criminologists. When handling missing data, the analytical techniques used in cross-national studies commonly rely on listwise deletion. With listwise deletion as the default in many longitudinal models, the analysis only includes waves for which there is complete coverage for all of the variables in the model, thereby limiting the total potential amount of information used for estimation.

To contend with missing data, we compared results from models using two different techniques: first, with listwise deletion as the default, we utilized Prais–Winsten regression models with panel corrected standard errors and an autoregressive (AR) (1) serial correlation; second, as an alternative to listwise deletion, we relied on full information maximum likelihood (FIML) as an option in linear, dynamic-panel data estimation, using maximum likelihood and structural equation modeling [10–12]. As an alternative to listwise deletion, FIML uses all the available information for estimation and does not drop observations with missing values.

## 2. Literature Review

As a predictor of homicide rates, we centered our analysis on urbanization, while controlling for other concepts that are theoretically related to homicide rates in the criminological literature, including age structure, economic productivity, labor conditions, educational attainment, and inequality [6,9,13–20]. We note that many of the variables presumed to influence homicide rates cross-nationally are also the same variables tested in cross-national studies by sustainability scholars, especially urbanization. Indeed, measures of urbanization have long been used as predictors of a wide variety of criminological and environmental outcomes [6,21]. In the literature review, we highlight recent research in criminology, which when synthesized presents urbanization as a multidimensional concept, a framework we derive from the sustainability literature.

### 2.1. The Multiple Dimensions of Urbanization

Social scientists have long noted that the commonly used basic measure of urbanization only establishes a lower threshold for identifying what is rural and urban and does not capture variation at the higher end of the urbanization continuum [22,23]. In other words, the basic measure of urbanization quantifies the overall proportion of the population living in cities (as opposed to rural areas), which establishes a general level of urbanization for that population. This measure is best used to highlight how a country's population is split between rural and urban areas; indeed, the notion of a rural-urban divide has long been a useful conceptual framework with a wide variety of cultural, practical, and theoretical implications [24]. However, this measure does not capture the degree of urban growth in terms of average city sizes adequately. Given the low threshold for the basic urban measure, two places with equal levels of urbanization can still exhibit very different degrees of urbanity. For instance, we observe that, in 2015, Greenland's population was about 86% urban, and in the United States that figure was roughly 82%. Nevertheless, in that year, the biggest city in Greenland had a population size less than 20,000. Thus, a singular measure based on the overall urban proportion does not sufficiently represent the scale of city life in the country.

Environmental social scientists have addressed these different dimensions, while noting that local-level and cross-national research had presented an apparent contradiction in the results for

urbanization: local-level scholars were finding that higher population density was associated with reduced environmental impacts; cross-national research was finding a positive association between urbanization and environmental impacts [2]. Indeed, looking at the quantitative research, the environmental consequences of these different dimensions of urbanization are not identical; they are unevenly associated with environmental change. As seen in the cross-national literature [25], much research has found a positive association between the general level of urbanization of a country and its per capita fossil fuel use. That is, compared to rural areas, urban areas have more productive and consumptive activities transpiring over a longer period of the day, which results in an increased use of fossil fuels. Meanwhile, much local-level research finds that high population densities can be negatively associated with per-capita fossil fuel use [26]. That finding supports the argument that greater density is associated with the more efficient deployment of a variety of technologies and infrastructure projects, including walkability, public transit, and residential high rises, that reduce fossil fuel use.

Meanwhile, in their analysis of county-level fossil fuel consumption in the United States, Elliott and Clement [2] clarify that these results are not contradictory; rather, these different levels of analysis are simply using different operational measures that represent separate dimensions of urbanization: one measure that captures the overall population split between rural and urban areas and another measure that captures average city size. In this way, environmental social scientists have framed urbanization as a multidimensional concept with divergent environmental impacts.

### 2.2. Urbanization versus City Size in Criminology

We evaluate this multidimensional framework of urbanization in the context of crime rates, with a specific focus on homicide rates. This is warranted because quantitative studies testing the relationship between urbanization and homicide show mixed results: several studies show a negative relationship between the proportion of the population residing in urban areas and homicide, contrary to criminological expectations discussed below, while others show a positive relationship [6,9]. Given the lengthy history of social science research on the relationship between urbanization and crime, herein we review, briefly, two competing frameworks on the topic: The first summarizes the traditional argument, still supported in cross-national research [6], that modern city life is criminogenic; the second draws on recent, local-level research on the non-monotonic results of urban scaling [4–7]. When taken together, these studies reaffirm that urbanization should be treated as a multidimensional concept, which in a cross-national study would involve capturing not only the overall urban proportion in the country but also an operational way to measure the presence of big cities in a country.

For the first framework, we note that there is a vast literature describing the criminogenic features of modern urban life [6,13]. Urbanization has been consistently hypothesized to increase crime rates, and homicide rates in particular. As discussed by Chamlin and Cochran [13], social control, structuralist, and subcultural theories contend that increases in population size are associated with increases in crime. Respectively, urbanization and population growth can weaken informal social control, can increase the likelihood that individuals come into contact with criminal individuals, and increase the expansion of deviant subcultures [13]. Criminologists have also applied theories of development to crime. Modernization theorists predicted that economic growth and rapid urban growth associated with 'modernization' (countries transitioning from rural economies to urban and capitalist economies) would lead to increases in crime because of limited economic opportunities, social dislocation, anomie, and low levels of informal social control [27,28]. In contrast, dependency theory posited that inequality and political repression would lead to more poor and marginalized people forced to internally migrate to cities to be criminalized in justice systems that are artifacts of colonial rule, thus raising official crime rates [29].

The second framework we reviewed draws from recent, local-level criminological work looking at urban-scale advantages [4,5,7], which is part of a broader quantitative literature that estimates the scaling relationships between city size and a variety of human behavioral and social science variables. Conceptually, the urban scaling literature provides insight into the agglomeration effects of urban

growth, asking whether there are economies of scale in city size. Analytically, this research estimates the percentage change in a particular outcome variable for every one percent increase in the population size of a city. For the estimation, the scaling literature computes the natural logarithm of both the predictor variable and outcome variable, yielding a scaling exponent for the slope estimate. A scaling exponent greater than 1 represents a superlinear relationship; an exponent less than 1 indicates a sublinear relationship; and a linear relationship is when the exponent is equal to 1. Moreover, given that both variables have been logged, the convention is to interpret the slope estimate as roughly equivalent to percent change in the dependent variable for every one percent change in the predictor.

The results from these studies show that some outcomes change at the same pace as changes in city size, while others grow at either slower or faster rates than the pace of city growth. Indeed, the topic of urban advantage and economies of scale is a central theme in the sustainability literature [2,3], where scholars and policymakers are concerned about improving the efficiency of resource use in urban areas. Meanwhile, in robust longitudinal studies, criminologists have recently applied urban scaling analysis to evaluate the potential for agglomeration effects in terms of homicide. For instance, using bivariate and multivariate longitudinal models in a study of US cities between 1999–2014, Chang et al. [4] found that the rates of both violent and property crimes were lower in bigger than in smaller cities. This result was also seen in a study of Indian cities [7], suggesting that there is an urban scale advantage in terms of violent crimes. Based on this second framework, we consider city size as a dimension of urbanization separate from the commonly used metric to identify the overall rural/urban divide.

## 2.3. Summary and Synthesis

As described above, studies on the relationship between crime and city size are conducted at the local-level; however, the present study takes a cross-national approach to evaluate the multidimensional framework of urbanization. Drawing from this literature review, our analysis examines two concepts that we measure at the country-level: a basic measure of urbanization to represent the overall rural-urban divide in a country and a separate measure that captures the proportion of the country's population living in large cities. While the analysis in our study focuses on homicide rates, we reiterate that quantitative sociologists in other areas use the same cross-national data sets in their analyses [3]. On that note, given the growing interest in the link between crime and sustainable development [4,8], we emphasize the relevance of our study for cross-national sustainability scholarship [3,30–33]. Like cross-national homicide studies, much of the quantitative environmental scholarship, with few exceptions [34,35], also handles missing values by way of listwise deletion, which we discuss in greater detail below. On that note, we now describe the data and analytic techniques used in our study.

## 3. Data and Analysis

Data used for this study came mostly from the World Bank's [36] online World Development Indicators, covering the years 2000–2015 for 217 countries (see Appendix A for list of countries). We also drew from the Standardized World Income Inequality Database [37] to access a measure of inequality (see below).

The World Development Indicators are regularly used by quantitative scholars studying the many dimensions of development [3,11,20,30–33,38,39]. Nevertheless, in the World Bank data, there are many countries that do not have complete longitudinal coverage for the variables being incorporated into the analysis. Therefore, given the panel estimation techniques commonly used in these studies (e.g., Prais–Winsten), if the longitudinal coverage is incomplete for a particular country (i.e., if a country has a single missing value for one variable in one time period), the wave with incomplete data for that country is entirely dropped from the model. As a result, with these particular panel techniques, some countries have more time periods of data than other countries, resulting in what are called unbalanced panels [33]. In this sense, the default for handling missing data in these commonly used longitudinal models is listwise deletion, which can result in a reduction in the overall sample size and the amount

of information used for estimation. In our study, we report results from Prais–Winsten models using the default of listwise deletion.

Some quantitative scholars [25,33,38] ran longitudinal models after having selected a stratified sample of countries using different criteria (e.g., countries with populations greater than 500,000 or one million; high income countries; least developed countries; Organization for Economic Co-Operation and Development (OECD) Countries; former Soviet republics, etc.). Nevertheless, as seen in Appendix B, many countries meeting these selection criteria, including OECD countries and countries with more than one million people, still have missing values, and with listwise deletion, get dropped from the panel models.

While some social scientists have utilized advanced imputation procedures (e.g., multiple imputation) to handle missing values [34,35], these approaches require the researcher to make numerous decisions affected by uncertainty [40]. As such, statisticians have cautioned against their use [40,41], now preferring instead readily available, user-friendly techniques using maximum likelihood for handling missing values in longitudinal data analysis. For instance, we considered the recent statistical work on linear, dynamic panel-data estimation using maximum likelihood and structural equation modeling [10–12].

For linear, dynamic panel-data estimation using maximum likelihood and SEM, Allison et al. [10] provide commands for use in a variety of statistical software packages. For our analysis, we utilized the command xtdpdml in Stata, the syntax of which is in Appendix C, which also provides the syntax for Prais-Winsten regression models with panel corrected standard errors and an AR (1) serial correlation, using the Stata command xtpcse.

Using structural equation modeling (SEM) for longitudinal data analysis allows the option to handle missing values with full information maximum likelihood (FIML). Here we provide a conceptual summary of this technique. With listwise deletion, if a case has a missing value in one time period for one variable but not another variable, that wave of information is still dropped from the longitudinal model. Moreover, with FIML, missing values are not imputed, as they are in mean imputation, linear imputation, or multiple imputation; instead, with FIML, the missing values are mathematically integrated out of the likelihood function; for a detailed review of the mathematics of this technique, see [10–12]. Thus, for the time period with partially complete coverage, the available information for that case is used to estimate the slope of the variable in the model. FIML has long been a common technique used to handle missing values in SEM [41,42], and criminologists have begun using this statistical tool to study the longitudinal variation in homicide rates across cities in the United States [43]. Compared to multiple imputation, the FIML option greatly simplifies the process of handling missing values in longitudinal models [40].

While we emphasize the benefits of full information maximum likelihood, we also acknowledge some of its limitations, particularly with respect to problems of computational speed and convergence [12]. Common data issues, such as high collinearity over time and severely unbalanced panel data, can slow down processing speed and prevent convergence. Indeed, linear dynamic panel-data estimation using maximum likelihood and structural equation modeling is a computationally intensive command and is sensitive to model specification and variable selection. Nevertheless, it has many options, and there are a number of other techniques available in various software packages to address these issues [12].

### 3.1. Variables

Table 1 displays all the variables used in our study, including their coverage and univariate summary statistics. Rather than the use of latent factors, as seen in some studies [9], the variables described below are direct measures of the primary concept of interest. The dependent variable is *homicide rate*, represented as the number of intentional homicides per 100,000 people for each country, derived by the World Bank from the United Nations Office on Drugs and Crime International Homicide Statistics Database. Including the lagged dependent variable, there are seven other predictor variables included in the analysis, the selection of which is based on commonly used measures in the cross-national literature.

**Table 1.** Univariate summary statistics (unlogged values).

| | 2000 | | | 2005 | | | 2010 | | | 2015 | | |
|---|---|---|---|---|---|---|---|---|---|---|---|---|
| | *n* | Mean | SD | *n* | Mean | SD | *n* | Mean | SD | *n* | Mean | SD |
| Homicide Rate | 145 | 8.469 | 10.560 | 174 | 8.021 | 10.515 | 188 | 8.311 | 11.844 | 137 | 7.444 | 12.647 |
| Proportion Urban | 215 | 55.491 | 24.613 | 215 | 56.968 | 24.377 | 215 | 58.479 | 24.264 | 214 | 60.025 | 24.114 |
| Proportion living in Large Cities | 121 | 23.591 | 17.096 | 121 | 24.274 | 17.286 | 121 | 24.665 | 17.196 | 121 | 25.328 | 17.388 |
| Proportion Male 15–24 | 194 | 9.676 | 1.908 | 194 | 9.620 | 2.040 | 194 | 9.135 | 2.051 | 193 | 8.532 | 2.265 |
| Proportion Unemployed | 187 | 8.890 | 6.188 | 187 | 8.588 | 6.341 | 187 | 8.496 | 6.032 | 187 | 8.021 | 6.084 |
| GDP (Purchasing Power Parity $ US) | 189 | 15,330.970 | 19,101.120 | 191 | 16,987.670 | 20,255.890 | 193 | 17,957.760 | 20,159.080 | 191 | 18,745.220 | 20,159.220 |
| Gini (Post-tax disposable income) | 150 | 38.914 | 8.229 | 169 | 39.077 | 7.790 | 162 | 38.224 | 7.523 | 111 | 37.140 | 7.900 |
| School Enrollment (Gender Parity Index) | 104 | 1.049 | 0.444 | 115 | 1.130 | 0.522 | 128 | 1.165 | 0.596 | 123 | 1.254 | 0.645 |

Note: All variables except GINI came from the World Development Indicators. The GINI variable came from the Standardized World Income Inequality Database (Solt 2016). Before including in panel models, the values for homicide rate and GDP were logged. Since there are some countries that have "0" homicides (not as missing values), a constant of "1" was added before taking the natural logarithm.

Urbanization variables included *proportion urban* (percentage of total population from the UN Population Division) and *proportion living in large cities*, defined as the proportion of the population in urban agglomerations of more than one million. Given that the *proportion urban* variable came from the commonly used UN data set, we recognize that each country tends to have a low threshold for what they define as urban (see United Nations 2005). As such, we also included the second variable *proportion living in large cities*, which helped to capture variation on the higher end of the urbanization spectrum.

Control variables included *proportion male 15–24*, *proportion unemployed*, *Gross Domestic Product (GDP)*, *Gini* (post-tax disposable income), and *tertiary (higher education) school enrollment*. *Proportion male 15–24* was calculated by adding together two variables: population ages 15–19, male (percentage of population that was male); and population ages 20–24, male (percentage of population that was male). Both of these variables were estimated based on age/sex distributions of the UN Population Division's World Population Prospects. *Proportion unemployed* (percentage of total labor force) was estimated by the International Labor Organization and came from the International Labor Organization Statistics (ILOSTAT) database. Economic productivity was measured by purchasing power parity (GDP) in current international (U.S.) dollars, from the World Bank's International Comparison Program database. The values of the GDP variable were log-transformed. The *Gini* coefficient came from the Standardized World Income Inequality Database (Version 6.2, March 2018) [37], which provides a measure of inequality in disposable (post-tax, post-transfer) income. Economic inequality is a consistent predictor of homicide in the literature [15,44]. The Gini coefficient or index is the most used measure of economic deprivation or inequality in cross-national studies on homicide rates, and the majority of studies find that Gini is positively related to homicide [9], though there are recent exceptions [18,19]. The rest of the independent variables came from the World Bank's online World Development Indicators. *Tertiary school enrollment* (gross gender parity index) is the ratio of women to men enrolled in public and private colleges and universities. The World Bank acquires the data from the UNESCO Institute for Statistics. The gender parity index of the gross enrollment ratio for each level of education was used in order to standardize the effects of the population structure of the appropriate age groups. A score closer to 1 means that a country is achieving equality between males and females in terms of access to higher education, while a score less than one favors males and a score greater than one favors females.

While we incorporated these control variables into our regression models, we acknowledge that other variables are theoretically relevant for cross-national homicide research (e.g., poverty, corruption in the public sector, and business-friendly regulations). However, when we incorporated these variables into the SEMs with the FIML option, the maximum likelihood estimation procedure did not converge. We discuss this issue below.

*3.2. Analysis Plan*

Below, we show the results of five panel models. The estimates in Models 1–2 are based on Prais–Winsten regression models with panel-corrected standard errors and an AR(1) serial correlation. With the FIML option, Models 3–5 are based on linear dynamic panel-data estimation using maximum likelihood and structural equation modeling. Model 3 does not include a lagged dependent variable; Models 4–5 do include a lagged dependent variable. All five models (1–5) incorporate fixed effects for units and time. While the use of fixed effects (including two-way fixed effects) is a common and recommended strategy for minimizing omitted variable bias in longitudinal analyses [45], we acknowledge the limitations of this technique, especially in terms of the interpretation of slope estimates. The syntax we used for estimating these models is reported in Appendix C. The generic equation for estimating all five models is the following:

$$y_{it} = \alpha_i + \gamma_t + x_{itk}b_k + \varepsilon_{it}, \tag{1}$$

wherein $y_{it}$ is the value of the dependent variable (the log-transformed homicide rate) for the i-th country at year t; $\alpha_i$ is the fixed effect for unit; $\gamma_t$ is the fixed effect for time; $x_{itk}$ equals the value of the

$k^{th}$ predictor for the i-th country at time t; $b_k$ represents the association between the k-th predictor and the dependent variable; and $\varepsilon_{it}$ is the country-specific error term at time t. In the Prais–Winsten regression models (Models 1–2), the error term $\varepsilon$ contains an estimate for $\rho$, the temporal autocorrelation parameter for the dependent variable. As a separate predictor k, Models 4–5 also have a slope estimate for $y_{t-1}$, the lagged dependent variable.

## 4. Results

The results from the panel models are displayed in Table 2. We organized the models to highlight different approaches to missing values. Models 1–2 relied on the default of listwise deletion, and Models 3–5 utilized the FIML option in SEM.

With a significance threshold of $p < 0.05$, Models 1 and 2 display similar results, with Model 2 including two additional control variables, including GINI and tertiary school enrollment, which reduced the sample size from $n = 113$ to $n = 93$. As seen in Appendix C, several of the countries excluded by listwise deletion are Organization for Economic Co-operation and Development (OECD) member nations and/or have populations greater than one million, which are commonly used criteria for stratification by quantitative scholars [9]. For instance, as the criterion for sample selection in their study of cross-national homicide rates, Kamprad and Liem [38] used a population size of 500,000, which yielded a sample size of $n = 165$. In Appendix C, we identify countries with a population of at least one million to demonstrate that a more conservative threshold still excludes a sizeable portion of the world's countries from panel analysis.

With that in mind, the results from Models 1–2 indicate that a country's homicide rate is positively associated with the proportion of the population who live in urban areas and who are young males and negatively associated with the proportion of urban residents who live in big cities and tertiary school enrollment. The slope estimates for GDP and unemployment are only marginally significant ($p < 0.1$).

Here we make two clarifying comments about the results in Table 2. First, we report the cross-sectional, pairwise correlations between the two urbanization variables for the four time periods ($p < 0.001$): 2000 = 0.6829; 2005 = 0.6712; 2010 = 0.6653; 2015 = 0.6530. Additionally, to check for multicollinearity, we ran an OLS model for each year, and from these models the maximum VIF was 6.87, which is below the threshold of concern for multicollinearity. Second, the high $R^2$ values for Models 1-2 are the result of including fixed effects for unit and time; we report the $R^2$ for the listwise deletion models simply to demonstrate that Model 2 has a higher $R^2$, and thus a better fit than Model 1.

Moving on to the FIML results, we first note that the sample size is the same in Models 3–5, regardless of which variables are added or whether a lagged dependent variable is incorporated into the model. As a direct comparison to Model 1, which is based on the default of listwise deletion from the Prais–Winsten approach, Model 3 has the same variables but uses FIML, which increases the sample size to $n = 216$ countries. In this model, the results for the urban variables are similar: homicide rates are positively associated with basic urbanization and negatively associated with the proportion living in big cities, a finding that is also seen in Models 4–5 after including a lagged dependent variable. However, with FIML, in Models 3–5, we highlight that the slope estimates for tertiary school enrollment and the proportion of the population who are young males are no longer significant. Meanwhile, looking at the fit statistics for the FIML models, we observe that Model 4 is the best fitting model, with the lowest $\chi^2$, RMSEA, and BIC, and the highest TFI and CLI. Model 5, with a significant slope estimate for the lagged dependent variable, still displays good fit and suggests that the only significant association with homicide rate comes from the proportion of urban residents who live in big cities. Indeed, this variable is the only variable to have a significant slope estimate in all five models, indicating that it is a robust finding, which we explore below.

**Table 2.** Cross-national, longitudinal analysis of the predictors of homicide rates, 2000–2015.

| Independent Variable | Model 1 Listwise Deletion | | | Model 2 Listwise Deletion | | | Model 3 FIML | | | Model 4 FIML | | | Model 5 FIML | | |
|---|---|---|---|---|---|---|---|---|---|---|---|---|---|---|---|
| | b | | SE | b | | SE | b | | SE | b | | SE | b | | SE |
| Homicide Lagged | | | | | | | | | | 0.139 | | 0.086 | 0.285 | ** | 0.103 |
| Proportion Urban | 0.016 | ** | 0.005 | 0.017 | ** | 0.006 | 0.024 | *** | 0.006 | 0.020 | * | 0.009 | 0.015 | | 0.010 |
| Proportion living in Large Cities | −0.033 | *** | 0.007 | −0.064 | *** | 0.009 | −0.088 | *** | 0.008 | −0.049 | ** | 0.015 | −0.029 | * | 0.014 |
| Proportion Male 15–24 | 0.049 | ** | 0.015 | 0.060 | ** | 0.018 | 0.018 | | 0.020 | 0.040 | | 0.024 | 0.035 | | 0.028 |
| Proportion Unemployed | 0.004 | | 0.004 | 0.009 | † | 0.005 | −0.009 | | 0.006 | −0.009 | | 0.008 | −0.013 | | 0.009 |
| GDP (Purchasing Power Parity $ US) | −0.114 | † | 0.060 | −0.087 | | 0.110 | −0.136 | | 0.091 | 0.002 | | 0.143 | −0.122 | | 0.155 |
| GINI (Post-tax disposable income) | | | | −0.013 | | 0.009 | | | | | | | 0.006 | | 0.015 |
| School Enrollment (Gender Parity Index) | | | | −0.258 | * | 0.128 | | | | | | | −0.017 | | 0.090 |
| R2 | 0.983 | | | 0.991 | | | | | | | | | | | |
| χ2 | | | | | | | 148.840 | *** | | 40.146 | | | 69.085 | ** | |
| RMSEA | | | | | | | 0.079 | | | 0.040 | | | 0.058 | | |
| CFI | | | | | | | 0.925 | | | 0.990 | | | 0.973 | | |
| TLI | | | | | | | 0.897 | | | 0.983 | | | 0.953 | | |
| BIC | | | | | | | 13,441.087 | | | 11,084.451 | | | 13,605.351 | | |
| Countries | 113 | | | 93 | | | 216 | | | 216 | | | 216 | | |
| Years | 4 | | | 4 | | | 4 | | | 4 | | | 4 | | |

† $p < 0.1$; * $p < 0.05$; ** $p < 0.01$; *** $p < 0.001$ (two-tailed tests). FIML = full information maximum likelihood.

## 5. Discussion and Conclusion

In this paper, drawing from the sustainability literature and synthesizing new criminological research, we evaluated urbanization as a multidimensional concept with countervailing criminological consequences. To that end, we collected country-level information from commonly-used longitudinal data sets, focusing on two different measures of urbanization: proportion of the population living in urban areas and proportion of the population living in cities with more than one million residents. Also in the analysis, we highlighted the issue of missing values in longitudinal, cross-national research, comparing results from models based on the default of listwise deletion and results from longitudinal structural equation modeling using the full information maximum likelihood (FIML) option, a technique that local-level criminologists have already begun to embrace [41]. For a cross-national study on homicide rates, utilizing the FIML option yielded a far higher sample size ($n = 216$) than what has been analyzed in previous scholarship [9]. Without FIML, the common default of listwise deletion in Model 1 ($n = 113$) excluded three OECD countries and 38 countries with populations of over one million, which is normal with the common criteria used for sample selection in the quantitative literature (see Appendix B). To be clear, in the social sciences, generally, the issue of missing data continues to be handled by way of listwise deletion or largely ignored, even in studies published in social science journals with an explicit focus on innovative empirical research [46–49].

In other words, for longitudinal cross-national analyses, stratified sampling does not guarantee complete coverage and only partially resolves the problem of missing data.

In the FIML models, aside from the lagged dependent variable, the only predictor variables significantly related to homicide were the two urbanization variables: proportion of the population that is urban and proportion of the population residing in urban agglomerations of over one million. In these models, we observed that the basic urbanization measure is positively associated with homicide; in other words, regardless of the typical size of the country's cities, as a country's population increasingly resides in urban areas its homicide rate also goes up. This finding corroborates Levchak's [6] robust analysis of the association between homicide and urbanization, which supports conventional theories on crime, including from both the modernization and dependency perspectives [27–29].

However, this basic measure of urbanization does not capture variation at the higher end of the urbanization continuum. Therefore, considering more recent, local-level longitudinal studies on homicide which examine city size [4,7], we incorporated a variable to measure the proportion of the population living in cities of one million or more. Indeed, the negative slope estimate for this variable corroborates results from these local-level studies, which showed that larger cities have lower rates of homicide than smaller cities. This finding stands in contrast to many theories of crime, such as subcultural, structuralist, and social control theories, which contend that population growth and urbanization would increase crime rates [13]. Instead, our findings suggest that there is a benefit of increased safety, at least in terms of homicide, as a country's population resides in bigger cities.

Also, we note that the technique for handling missing values, either by listwise deletion or FIML, influenced the findings for the control variables. In fact, when using FIML, with the exception of the lagged dependent variable, not a single slope estimate for any of the control variables was statistically significant. In our study, while we treated these variables as important controls, their nonsignificant estimates are noteworthy, considering that all of these variables are theoretically relevant, and previous studies using listwise deletion have been observed them as significant predictors of homicide [9]. Conversely, in terms of the urbanization variables, the technique for handling missing values did not matter. No matter the sample size, the significance and direction of the slope estimate for the large cities variable was consistent in all five models, indicating a robust cross-national relationship with homicides.

Indeed, given that the large cities variable was the only variable to be significant in all models, this finding lays the groundwork for future criminological research to explore the multifaceted relationship between urbanization and homicide. While we controlled for official unemployment rates, perhaps larger cities have advantages when it comes to reducing homicide risk, such as presenting

more informal economic opportunities, and more developed security sectors, including having more advanced technologies that enhance surveillance and control. Whatever the mechanism, this study presents evidence of an urban scale advantage for homicides. Additionally, in light of the relevance of urbanization for sustainability research [2,3,25], we also outlined this finding as a constructive piece of the emerging literature on the link between crime and sustainability [8]. Indeed, the concepts behind all the predictor variables used in our analysis play important roles in a range of theories on development and sustainability; therefore, to improve estimation and generalizability, future cross-national research on these topics can utilize SEM with the FIML option.

While the results from this study outline an urbanization framework for future criminological research to consider, we acknowledge three limitations in terms of our analytic approach. First, as already described in the data section, the FIML technique is computationally intensive and may generate problems of convergence in estimation. While we focused on theoretically relevant control variables, in supplemental analyses, as discussed above, we confronted problems of convergence when incorporating additional variables to control for levels of poverty, corruption, and business-friendly regulations. Future scholarship could explore whether this problem happens with other theoretically relevant variables that are included in the analysis. Second, while we found robust evidence that the large cities variable is negatively related to homicides, this was an average relationship for all countries; we did not test for spatial heterogeneity in the slope estimate; i.e., whether the slope estimate for the predictor varied across space. Drawing from other urban scaling research [5], rather than estimating an average association, future scholars can consider whether the slope estimate for the large cities predictor varies from region to region (or even country to country). Third, in a similar manner, we also did not test whether there was variation in the slope estimate over time. As Chang et al. [6] discussed, in their study of crime rates in US cities, the degree and timing of change in homicide rates varied by city size, and large cities experienced a more noticeable decline in the 21st century. On that note, similar to the question of spatial variation, cross-national criminologists can also explore the issue of temporal variation in the relationship between city size and homicide rates.

**Author Contributions:** Conceptualization: M.T.C., N.W.P., and J.B.; Data Collection: M.T.C. and N.W.P.; Statistical Analysis: M.T.C.; Writing: M.T.C., N.W.P., and J.B.

**Funding:** This research received no external funding.

**Conflicts of Interest:** The authors declare no conflicts of interest.

## Appendix A

**Table A1.** List of countries included in the study (*n* = 217) *.

| | | |
|---|---|---|
| Afghanistan | Albania | Algeria |
| American Samoa | Andorra | Angola |
| Antigua and Barbuda | Argentina | Armenia |
| Aruba | Australia | Austria |
| Azerbaijan | Bahamas, The | Bahrain |
| Bangladesh | Barbados | Belarus |
| Belgium | Belize | Benin |
| Bermuda | Bhutan | Bolivia |
| Bosnia and Herzegovina | Botswana | Brazil |
| British Virgin Islands | Brunei Darussalam | Bulgaria |
| Burkina Faso | Burundi | Cabo Verde |
| Cambodia | Cameroon | Canada |
| Cayman Islands | Central African Republic | Chad |
| Channel Islands | Chile | China |
| Colombia | Comoros | Congo, Dem. Rep. |
| Congo, Rep. | Costa Rica | Cote d'Ivoire |
| Croatia | Cuba | Curacao |
| Cyprus | Czech Republic | Denmark |

**Table A1.** *Cont.*

| | | |
|---|---|---|
| Djibouti | Dominica | Dominican Republic |
| Ecuador | Egypt, Arab Rep. | El Salvador |
| Equatorial Guinea | Eritrea | Estonia |
| Eswatini | Ethiopia | Faroe Islands |
| Fiji | Finland | France |
| French Polynesia | Gabon | Gambia, The |
| Georgia | Germany | Ghana |
| Gibraltar | Greece | Greenland |
| Grenada | Guam | Guatemala |
| Guinea | Guinea-Bissau | Guyana |
| Haiti | Honduras | Hong Kong SAR, China |
| Hungary | Iceland | India |
| Indonesia | Iran, Islamic Rep. | Iraq |
| Ireland | Isle of Man | Israel |
| Italy | Jamaica | Japan |
| Jordan | Kazakhstan | Kenya |
| Kiribati | Korea, Dem. People's Rep. | Korea, Rep. |
| Kosovo | Kuwait | Kyrgyz Republic |
| Lao PDR | Latvia | Lebanon |
| Lesotho | Liberia | Libya |
| Liechtenstein | Lithuania | Luxembourg |
| Macao SAR, China | Macedonia, FYR | Madagascar |
| Malawi | Malaysia | Maldives |
| Mali | Malta | Marshall Islands |
| Mauritania | Mauritius | Mexico |
| Micronesia, Fed. Sts. | Moldova | Monaco |
| Mongolia | Montenegro | Morocco |
| Mozambique | Myanmar | Namibia |
| Nauru | Nepal | Netherlands |
| New Caledonia | New Zealand | Nicaragua |
| Niger | Nigeria | Northern Mariana Islands |
| Norway | Oman | Pakistan |
| Palau | Panama | Papua New Guinea |
| Paraguay | Peru | Philippines |
| Poland | Portugal | Puerto Rico |
| Qatar | Romania | Russian Federation |
| Rwanda | Samoa | San Marino |
| Sao Tome and Principe | Saudi Arabia | Senegal |
| Serbia | Seychelles | Sierra Leone |
| Singapore | Sint Maarten (Dutch part) | Slovak Republic |
| Slovenia | Solomon Islands | Somalia |
| South Africa | South Sudan | Spain |
| Sri Lanka | St. Kitts and Nevis | St. Lucia |
| St. Martin (French part) * | Vincent and the Grenadines | Sudan |
| Suriname | Sweden | Switzerland |
| Syrian Arab Republic | Tajikistan | Tanzania |
| Thailand | Timor-Leste | Togo |
| Tonga | Trinidad and Tobago | Tunisia |
| Turkey | Turkmenistan | Turks and Caicos Islands |
| Tuvalu | Uganda | Ukraine |
| United Arab Emirates | United Kingdom | United States |
| Uruguay | Uzbekistan | Vanuatu |
| Venezuela, RB | Vietnam | Virgin Islands (U.S.) |
| West Bank and Gaza | Yemen, Rep. | Zambia |
| Zimbabwe | | |

* Note: Only one country, St. Martin (French Part), had missing values for all variables and all waves; this country is not included in the xtdpdml model using full information maximum likelihood for missing values. This yields a final *n* = 216 for the xtdpdml models.

## Appendix B

**Table A2.** List of countries with missing values excluded from Model 1 using xtpcse (*n* = 104) *.

| | |
|---|---|
| Albania ‡ | American Samoa |
| Andorra | Angola ‡ |
| Antigua and Barbuda | Aruba |
| Bahamas, The | Bahrain ‡ |
| Barbados | Belize |
| Benin ‡ | Bermuda |
| Bhutan | Bosnia and Herzegovina |
| Botswana | British Virgin Islands |
| Brunei Darussalam | Burundi |
| Cabo Verde | Cayman Islands |
| Central African Republic ‡ | Channel Islands |
| Comoros | Croatia |
| Cuba ‡ | Curacao |
| Cyprus ‡ | Djibouti |
| Dominica | Equatorial Guinea ‡ |
| Eritrea | Estonia †,‡ |
| Eswatini ‡ | Faroe Islands |
| Fiji | French Polynesia |
| Gabon ‡ | Gambia, The ‡ |
| Gibraltar | Greenland |
| Grenada | Guam |
| Guinea-Bissau ‡ | Guyana |
| Iceland † | Iran, Islamic Rep. ‡ |
| Isle of Man | Jamaica ‡ |
| Kiribati | Korea, Dem. People's Rep. ‡ |
| Kosovo ‡ | Kyrgyz Republic ‡ |
| Lao PDR ‡ | Latvia ‡ |
| Lesotho ‡ | Liechtenstein |
| Lithuania ‡ | Luxembourg † |
| Macao SAR, China | Macedonia, FYR |
| Maldives | Malta |
| Marshall Islands | Mauritius ‡ |
| Micronesia, Fed. Sts. | Moldova ‡ |
| Monaco | Montenegro |
| Namibia ‡ | Nauru |
| New Caledonia | Niger ‡ |
| Northern Mariana Islands | Palau |
| Papua New Guinea ‡ | Qatar |
| Samoa | San Marino |
| Sao Tome and Principe | Seychelles |
| Sint Maarten (Dutch part) | Slovak Republic ‡ |
| Slovenia ‡ | Solomon Islands |
| Somalia ‡ | South Sudan ‡ |
| Sri Lanka ‡ | St. Kitts and Nevis |
| St. Lucia | St. Martin (French part) |
| Vincent and the Grenadines | Sudan ‡ |
| Suriname | Syrian Arab Republic ‡ |
| Tajikistan ‡ | Timor-Leste ‡ |
| Tonga | Trinidad and Tobago ‡ |
| Turkmenistan ‡ | Turks and Caicos Islands |
| Tuvalu | Vanuatu |
| Virgin Islands (U.S.) | West Bank and Gaza ‡ |

* Note: The sample size of countries included in Model 1 is *n* = 113. We use the Stata command xtpcse; like most commonly used panel estimation commands, xtpcse will drop any case that is missing at least one value of a variable in all waves. † Note: Country designated as an OECD member nation. ‡ Note: Country has a population greater than one million.



**Appendix C. Stata Commands Used to Generate Estimates in Table 2**

\*\* Variable names and descriptions\*\*
\* homicide = Homicide Rate Per 100,000 People;
\* urban = Proportion of Population Living in Cities;
\* megaurban = Proportion of Urban Population Living in Large Cities of one million or more;
\* youthmale = Proportion of Population who is Male, Aged 15-24;
\* unemptotilo = Proportion of Labor Force who is Unemployed;
\* gdpppp = GDP Per Capita (In 2010 $ US, Purchasing Power Parity);
\* swiid = Meause of GINI from the Standardized World Income Inequality Database;
\* schoolgpit = Tertiary School Enrollment (Gross, Gender Parity Index).

\*\* Set id and time variables \*\*
xtset id time

\*\* Recode time variable as appropriate for use with xtdpdml \*\*
replace time = 1 if time == 2000;
replace time = 2 if time == 2005;
replace time = 3 if time == 2010;
replace time = 4 if time == 2015.

\*\* Longitudinal model estimates \*\*

\* Models 1–2: Prais-Wintsten with two-way fixed effects, panel corrected standard errors, and AR1 correlation.

\* Model 1
xtpcse homicide urban megaurban youthmale unemptotilo gdpppp i.id i.time, corr(ar1) het.

\* Model 2, with two additional control variables
xtpcse homicide urban megaurban youthmale unemptotilo gdpppp swiid schoolgpit i.id i.time, corr(ar1) het.

\* Models 3–5: xtdpdml with two-way fixed effects.
\* ylag(0) = no lagged dependent variable.
\* errorinv = error variances same at each time period.
\* dif = difficult.
\* fiml = full information maximum likelihood; missing values with maximum likelihood.

\* Model 3, no lagged dependent variable.
xtdpdml homicide urban megaurban youthmale unemptotilo gdpppp, ylag(0) errorinv fiml dif.

\* Model 4, with lagged dependent variable.
xtdpdml homicide urban megaurban youthmale unemptotilo gdpppp, errorinv fiml dif.

\* Model 5, with lagged dependent variable and two additional controls.
xtdpdml homicide urban megaurban youthmale unemptotilo gdpppp swiid schoolgpit, errorinv fiml dif.

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
