# Peer review of "Homicide Rates and the Multiple Dimensions of Urbanization: A Longitudinal, Cross-National Analysis"

_sustainability, doi:10.3390/su11205855_

Round 1

Reviewer 1 Report

Dear Authors,

First of all I would like to congratulate for such a great piece of work. The aim of the study is a very interesting field in Criminology but I would say that the most appealing view of your research is the combination of the two different models to address the topic.

Longer review version:

"There is no doubt about the relationship between homicide rates and city’s development. In fact stablishing an addressing this relation has been a wide studied topic in the literature. Although this paper is about this relationship and its consequences there are several aspects that make it a very refreshing and new point of view of that. First of all because they recognize the limitations of the previous cross-national studies on homicide rates and stablish this as their starting point. So, they take under consideration the importance of the sample size and how it leads to missing data. According to this, to contend with missing data they use two different techniques, one as the default (listwise deletion) and the alternative (full information maximum likelihood).

One of the most important strengths of this paper is how they understand urbanization in order to associate it to homicide rates. So, they treat urbanization as a multidimensional concept with countervailing and non-monotonic consequences but they also include traditional controlling concepts used by the literature such as age, structure, economic productivity, labor conditions, educational attainment, and inequality. It is also important to highlight how this research takes a cross-national approach to address the issue, instead of the local-level approach whis has been the usual.

The data used come from two different resources: The World Bank’s online World Development Indicators and the Standardized World Income Inequality Database, which means the research has a wide overview of the data an reality from different ways and terms of measurement.

Regarding to the main findings, they noticed that for both models (listwise deletion and full information maximum likelihood) homicide rate is positively associated with basic urbanization (understood as the proportion of population who live in urban areas) and negatively associated with the proportion living in big cities. However, while in listwise deletion, tertiary school enrolment and young male proportion have revealed as a negatively associated indicators, with full information maximum likelihood these two indicators are no longer significant."

Author Response

We thank the reviewer for their comments. We will edit the manuscript to fix the minor spell checking issues noted by the reviewer.

Reviewer 2 Report

Summary

The purpose of this study is twofold. First, following sustainability scholars, the authors suggest that urbanization is multi-dimensional. To this end, they incorporate two measures of urbanization into a cross-national analysis of homicide rates. The first measure is the proportion of a nation’s residents who reside in urban areas. The second is the proportion living in large cities. Next, the authors argue that existing studies of cross-national homicide rates suffer from small sample sizes. To this end, the authors use two methods to analyze the association between the two urbanization measures and homicide. The first is Prais-Winston regression. The second is structural equation modeling with full information maximum likelihood estimation. The first method utilizes listwise deletion which reduces the number of nations in the analysis, while the second method preserves all nations from the full sample. The authors find that, regardless of method, both measures of urbanization are significantly associated with homicide.

Strengths

There are many strengths to this article. Most significantly, the authors draw on methods (i.e. structural equation modeling with full information maximum likelihood estimation) that allow them to retain a large sample size. This is especially important in the context of cross-national research where missing data are generally handled by dropping cases via listwise deletion, thereby producing sample sizes that are relatively small and, likely, unrepresentative. The authors’ comparison of Prais-Winston estimates to SEM FIML estimates displays the robust association between urbanization and homicide – a finding that others have also observed (e.g. Levchak 2016). The authors also find that the two measures of urbanization operate differently finding the level of urbanization to be positively associated with urbanization and the proportion in large cities to be negatively associated with homicide. This indicates that scholars of cross-national homicide would be well-advised to consider the multi-dimensionality of urbanization.

Weaknesses

There are also several weaknesses. While the authors should be commended for being able to include so many nations in the analysis, they choose to only include a small number of covariates. The authors cite several studies by Pridemore (2000, 2011, 2012), but make little mention of the research on the role of poverty (Pridemore 2008, 2011). There are also no measures of democracy (from Freedom House) or political terror (the Political Terror Scale) – which are commonly included in cross-national studies of homicide. I don’t suspect their inclusion would change any of the results, but I am curious why the authors did not include more covariates. This could be checked.

I would like to see the correlation between the two urbanization variables – proportion urban and proportion living in large cities. While they are producing different effects – positive and negative, respectively – there is no indication of how collinear they are. Additionally, the authors could examine the direction and magnitude of association when each is entered separately into each model. This could then be mentioned in a footnote.

I also believe the introduction should be reorganized. The manuscript is titled “Homicide Rates and the Multiple Dimensions of Urbanization: A Longitudinal Cross-National Analysis.” However, the introduction begins by focusing on the problem of missing data. Is the primary purpose of the manuscript to compare results obtained via Prais-Winston regression and those obtained via SEM FIML or is it to incorporate multiple dimensions into a longitudinal cross-national study of homicide? Of course, it can be both, but the introduction reads as a comparison of methodological approaches.

Specific Comments

There is a typo toward the end of page 12: “i.e., whether the slope estimate for the predictor varies space.”

Author Response

We thank the reviewer for their constructive comments and suggestions for improving the manuscript. Here we respond to two issues addressed in their review.

First, we agree with the reviewer the introduction should be edited to reflect our primary goal of assessing the multidimensionality of urbanization. On that note, we have revised the introduction accordingly, first discussing the multidimensional framework of urbanization and then highlighting the issue of missing values.

Second,we appreciate the reviewer's comment about the need to incorporate additional control variables into the model. Unfortunately, as we discuss in the manuscript, the SEM-FIML models are very sensitive to specification. For instance, in initial analyses using SEM-FIML, we had included in separate models two different variables for poverty from the World Development Indicators (% below $1.90 and % below national poverty lines). However, estimate convergence only happened when we had three variables in the model: percent urban, large cities, and poverty. While the large cities variable still had a significant, negative slope estimate, the fit statistics for this model were not good. The same thing happened for two other variables from the World Development Indicators: corruption in the public sector and business-friendly regulations. It should be noted these variables have many more missing values than the other variables we used from the World Bank. While we did not have access to the Freedom House variable in the World Bank data set, if the reviewer has suggestions on how to merge this with the World Bank data, we would be happy to incorporate this information in additional analyses.

All the same, to address these issues, we have now included the following footnote in the manuscript, under the section on control variables: "While we incorporate these control variables into our regression models, we acknowledge that there are other variables theoretically relevant for cross-national homicide research (e.g., poverty, corruption in the public sector, business-friendly regulations). However, when we incorporated these variables into the SEMs with the FIML option, the maximum likelihood estimation procedure did not converge. We discuss this issue below."  

Lastly, to address the reviewer's question about multicollinearity, we now incorporate the following as footnote in the results section:

"Here we report the cross-sectional correlations between the two urbanization variables for the four time periods (p<0.001): 2000 = 0.6829; 2005= 0.6712; 2010 = 0.6653; 2015 = 0.6530. Also, to check for multicollinearity, we ran an OLS model for each year, and from these models the maximum VIF was 6.87, which is below the threshold of concern for multicollinearity."